# Programming DNA origami patterning with non-canonical DNA-based metallization reactions

Sisi Jia [1,2,9], Jianbang Wang[2,9], Mo Xie[2], Jixue Sun[3], Huajie Liu[4*], Yinan Zhang[2], Jie Chao [5], Jiang Li[2,6], Lihua Wang[2,7], Jianping Lin[3], Kurt V. Gothelf [8] & Chunhai Fan [1*]

The inherent specificity of DNA sequence hybridization has been extensively exploited to develop bioengineering applications. Nevertheless, the structural potential of DNA has been far less explored for creating non-canonical DNA-based reactions. Here we develop a DNA origami-enabled highly localized metallization reaction for intrinsic metallization patterning with 10-nm resolution. Both theoretical and experimental studies reveal that low-valence metal ions ($Cu^{2+}$ and $Ag^+$) strongly coordinate with DNA bases in protruding clustered DNA (pcDNA) prescribed on two-dimensional DNA origami, which results in effective attraction within flexible pcDNA strands for site-specific pcDNA condensation. We find that the metallization reactions occur selectively on prescribed sites while not on origami substrates. This strategy is generically applicable for free-style metal painting of alphabet letters, digits and geometric shapes on all−DNA substrates with near-unity efficiency. We have further fabricated single- and double-layer nanoscale printed circuit board (nano-PCB) mimics, shedding light on bio-inspired fabrication for nanoelectronic and nanophotonic applications.

[1] School of Chemistry and Chemical Engineering, and Institute of Molecular Medicine, Renji Hospital, School of Medicine, Shanghai Jiao Tong University, Shanghai 200240, China. [2] Division of Physical Biology, CAS Key Laboratory of Interfacial Physics and Technology, Shanghai Synchrotron Radiation Facility, Shanghai Institute of Applied Physics, Chinese Academy of Sciences, Shanghai 201800, China. [3] State Key Laboratory of Medicinal Chemical Biology and College of Pharmacy, Nankai University, Tianjin 300071, China. [4] School of Chemical Science and Engineering, Shanghai Research Institute for Intelligent Autonomous Systems, Key Laboratory of Advanced Civil Engineering Materials of Ministry of Education, Tongji University, Shanghai 200092, China. [5] Key Laboratory for Organic Electronics & Information Displays (KLOEID), Institute of Advanced Materials (IAM) and School of Materials Science and Engineering, Nanjing University of Posts & Telecommunications, Nanjing 210046, China. [6] Zhangjiang Laboratory, Shanghai Advanced Research Institute, Chinese Academy of Sciences, Shanghai 201210, China. [7] Shanghai Key Laboratory of Green Chemistry and Chemical Processes, School of Chemistry and Molecular Engineering, East China Normal University, 500 Dongchuan Road, Shanghai 200241, China. [8] Center for DNA Nanotechnology (CDNA) at the Interdisciplinary Nanoscience Center (iNANO) and the Department of Chemistry, Aarhus University, 8000 Aarhus, Denmark. [9]These authors contributed equally: Sisi Jia, Jianbang Wang *email: liuhuajie@tongji.edu.cn; fanchunhai@sjtu.edu.cn

The fidelity of the genetics of living organisms is deeply rooted in the inherent Watson−Crick pairing specificity of DNA sequence hybridization reactions. Over the past several decades, researchers have been actively repurposing highly selective hybridization reactions for various engineering purposes including material construction, organic synthesis, drug screening and delivery and theranostics. Condensation is a less explored intrinsic property of DNA molecules as compared to DNA hybridization, possibly due to the lack of selectivity in dilute, noncrowded bulk solution. However, condensation of anionic DNA molecules by high valence cations ($z \geq 3$) is under programmed spatiotemporal control in viral, bacterial and mammalian cells[1,2]. For example, meter-scale genome DNA is compacted in the micrometer-scale nuclei with 10-nm scale ordered and ultrafine structural features in the crowded environment of mammalian cells[1,2]. We were inspired by the in vivo condensation that cations can induce negatively charged DNA chains to form compact structures. Here we employ two-dimensional (2D) DNA origami to program the compact of single-stranded (ss-) DNA in vitro.

DNA origami is well proven to have the power for custom-making precise nanostructures[3–12]. From a structural point of view, the periodic two-dimensional (2D) structure of origami resembles a 2D crystal[3], as compared to ssDNA. The introduction of exogenous nanoobjects on the origami crystal can in principle create noneven distribution of defect clusters for developing nanoscale soft lithography[13–17]. A typical approach involves the anchoring of discrete metal nanoparticle (NP) seeds on prescribed positions of origami, allowing site-specific metallization to form metal nanopatterns[18,19]. Nevertheless, the indispensable and error-prone NP anchoring makes it difficult to fabricate ultrafine and continuous patterns[18,19].

Here, we report the development of 2D DNA origami-enabled selective DNA condensation and metallization reactions free of exogenous metal NPs. Metal ions with low valence ($z < 3$) can efficiently and site-specifically condense protruding clustered DNA (pcDNA) prescribed on DNA origami. Interestingly, we find that metal plating occurs predominantly on these condensed sites, resulting in well-defined nanopatterns with 10-nm resolution, high line density and near-unity yields. The intrinsic and programmable nature of this free-style origami painting strategy enables the fabrication of nanoscale printed circuit boards (nano-PCB).

## Results

**The design principle**. We first designed a 2D rectangular DNA origami sheet with a size of $100 \times 70$ nm$^2$ and a thickness of 2 nm, which was made of a long scaffold M13mp18 viral DNA (7249 bases) and ~200 short staple strands (Fig. 1a and Supplementary Fig. 1). This origami was employed as the canvas to support addressable DNA condensation and intrinsic metallization patterning (DCIMP). A blueprint of the nanopattern (a digit 8) was prescribed on the origami (Supplementary Fig. 22). Technically, this procedure equals to the site-specific placing of pcDNA on the origami substrate by elongating selected staple strands on pre-defined positions. Each pcDNA site is composed of three single-stranded (ss-) DNA with 30 bases, which is seamlessly interfaced on the double-stranded (ds-) DNA-based origami substrate (dubbed osDNA). Of note, given that the bending stiffness of a given object is proportional to the fourth order of its effective radius, the bending stiffness of osDNA is several orders of magnitude higher than that of pcDNA[20]. Thus, we reason that metal ions-induced condensation of the negative ion atmosphere of DNA would selectively occur on flexible pcDNA rather than relatively stiff osDNA.

To test this designing principle for the DCIMP strategy, we employed copper ions (Cu$^{2+}$) to condense pcDNA and electroless plating to metalize condensed patterns. A freshly mixed solution of CuCl$_2$ (4 mM) and a reducing agent, ascorbic acid (20 mM), in a Tris-acetate-Mg$^{2+}$ (TA-Mg$^{2+}$) buffer was dropcast on mica to react with the origami. The copper plating was stopped by replacing the solution with fresh TA-Mg$^{2+}$ buffer after 10-min incubation at room temperature. Atomic force microscopy (AFM) characterization visualized the formed pattern of digit 8 on the prescribed pcDNA region (Supplementary Fig. 2). We found that the lines in the pattern had a mean height of ~4 nm from the mica, or ~2 nm from the origami. We did not find any morphological change on the osDNA region even after 20-min incubation, which strongly implied the extraordinarily high selectivity of condensation-based Cu metallization. Control studies revealed that metallization could not be initiated in the absence of either pcDNA or AscH/CuCl$_2$ (Supplementary Fig. 3), which further substantiated the pcDNA-over-osDNA selectivity for Cu metallization.

We next studied the time- and concentration-dependence of this condensation-based Cu metallization. Time-evolution experiments suggested that this on-origami metallization was complete within <1 min (Supplementary Fig. 4). Further increase in the reaction time up to 20 min did not result in apparent morphological change, suggesting that the metallization is self-limited. Also interestingly, concentration of Cu$^{2+}$ plays a critical role in metallization. Concentration profile (0.1–100 mM) studies showed a three-phase concentration-dependent metallization (Fig. 1b and Supplementary Fig. 5). In the low concentration range (0.1–1 mM), we observed partial metallization that led to discontinuous Cu dots; in the medium concentration range (2–10 mM), metallization was highly site-specific, producing continuous line nanopatterns of digit 8; in the high concentration range (20–100 mM), we found that lines were broadened, resulting in blurred nanopatterns. In addition, the whole origami substrate was nearly completely metalized in the presence of 100 mM Cu$^{2+}$ (Supplementary Fig. 6), suggesting the loss of selectivity at this very high concentration.

Tomographic analysis provides a quantitative approach to study this site-specific metallization (Fig. 1c). By examining tomography of cross-sections in AFM images at the heights of 1 nm (origami substrate) and 3 nm (metalized area), we observed clear digit 8 nanopatterns at optimal Cu$^{2+}$ concentrations. In contrast, insufficient and excess Cu$^{2+}$ led to either discontinuous spots or blurred images. Theoretically, metalized portion should occupy ~31–44% of the surface area of origami (see Eqs. 1, 2 and Supplementary Fig. 7 in the supplementary information). Indeed, we observed a transition of height change at ~40% occupancy on origami when the Cu$^{2+}$ concentration was less than 10 mM (left in Fig. 1d), implying high pcDNA-over-osDNA selectivity at low-to-medium concentrations of Cu$^{2+}$. The transition disappeared when Cu$^{2+}$ concentration was >20 mM, suggesting the loss of the selectivity at high concentrations. Especially, by putting 1-nm height increase as the threshold for effective metallization, we found that the optimal Cu$^{2+}$ concentrations were in the range of 2–10 mM (right in Fig. 1d), consistent with the morphological observation.

Having established selective condensation and metallization patterning on pcDNA sites of the origami, we next performed quantum mechanical calculations to explore the mechanism. We evaluated the binding affinities of Cu$^{2+}$ ions to DNA bases. As shown in Fig. 1e, Cu$^{2+}$ ions strongly coordinated with DNA bases via the N–Cu interactions to form Cu-pcDNA complexes, which accounts for the observed accumulation of Cu$^{2+}$ ions surrounding unpaired bases on pcDNA. Since Mg$^{2+}$ ions are used for the origami assembly, we next examined their competition binding

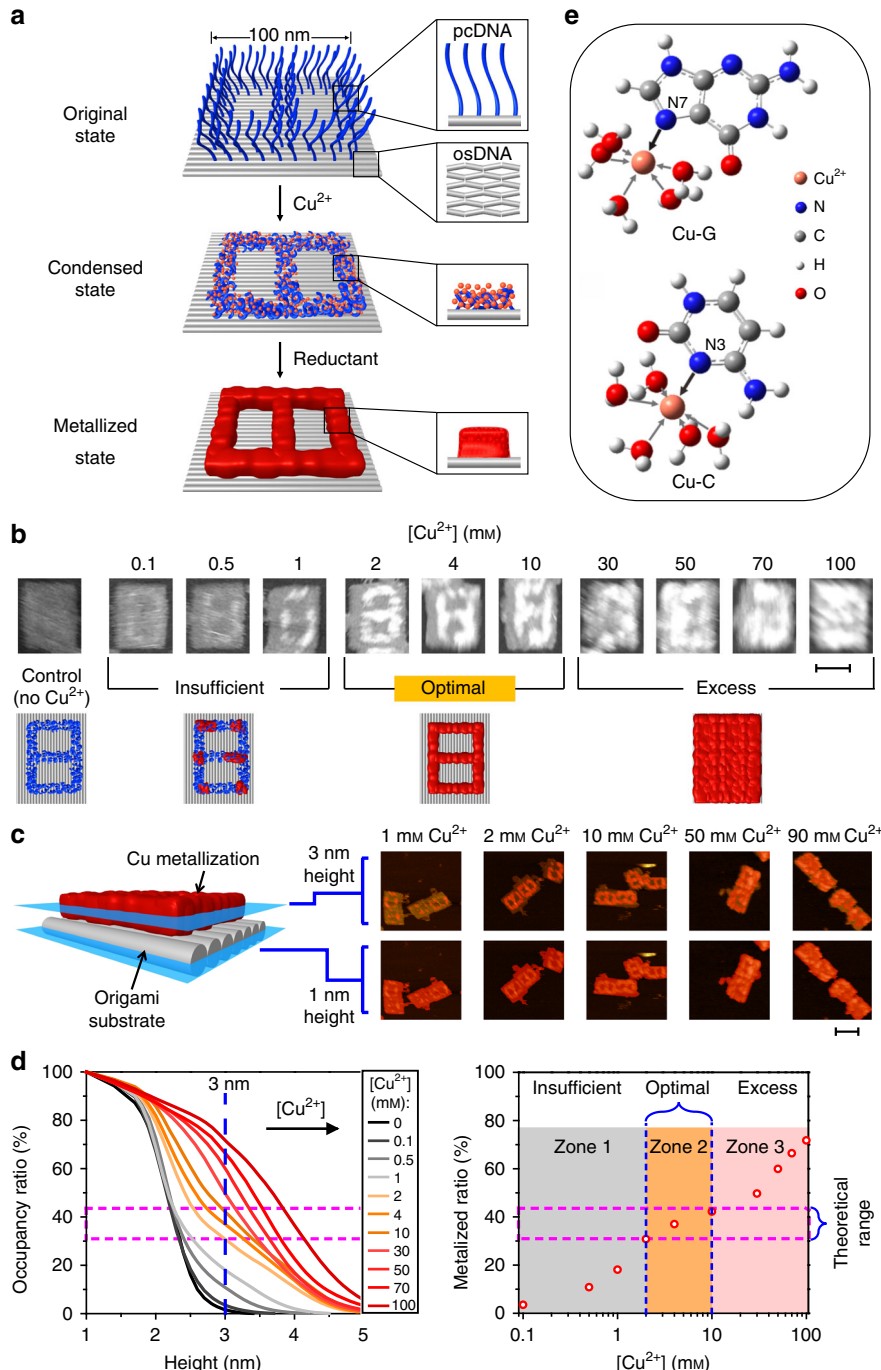

**Fig. 1** Selective on-origami DNA condensation and metallization. **a** Illustration of condensation and subsequent site-specific metal plating processes for fabricating a digit 8 pattern on a single-origami breadboard with a size of $100 \times 70$ nm$^2$. **b** AFM images of metalized digit 8 patterns constructed with different Cu$^{2+}$ concentrations. Scale bar: 50 nm. **c** Tomographic measurements of cross-sectional areas. Tomography images (showing red masks on original AFM images) of origami substrates (cross-sections at 1 nm height) and metallized areas (cross-sections at 3 nm height) from typical samples. Scale bar: 100 nm. **d** Normalized occupancy ratios of the areas on origami at different heights of cross-sections obtained from tomography analysis (left) of the patterns showing in (**b**). The size of the origami substrate (measured from the cross-sectional area at 1 nm—half the original origami height) is represented with 100%. Metallized area proportions with different Cu$^{2+}$ concentrations obtained from the normalized occupancy data at 3 nm height (1 nm above from the origami substrate) (right). **e** Geometry-minimized structure of Cu-G and Cu-C complexes obtained from quantum mechanical calculations.

by comparing the affinity of Mg$^{2+}$ and Cu$^{2+}$ to pcDNA. Importantly, Cu$^{2+}$ ions showed ~10 kcal mol$^{-1}$ higher affinity than Mg$^{2+}$ ions when they bind to bases, regardless of G, C or A ($-18.26$ vs. $-8.69$, $-15.58$ vs. $-5.44$, $-10.74$ vs. $-0.54$, with units of kcal mol$^{-1}$). Given that DNA bases are more exposed in flexible pcDNA than in relatively stiff osDNA, the high local density of Cu$^{2+}$ overcomes the intra-strand repulsion and

induces the attraction of pcDNA, resulting in site-specific condensation on DNA origami. As a consequence, pcDNA acts as the nucleation site to induce Cu metallization and growth in the presence of AscH.

**Selectivity and features of DCIMP.** Next, we studied the number and the length effects of pcDNA on the selectivity of

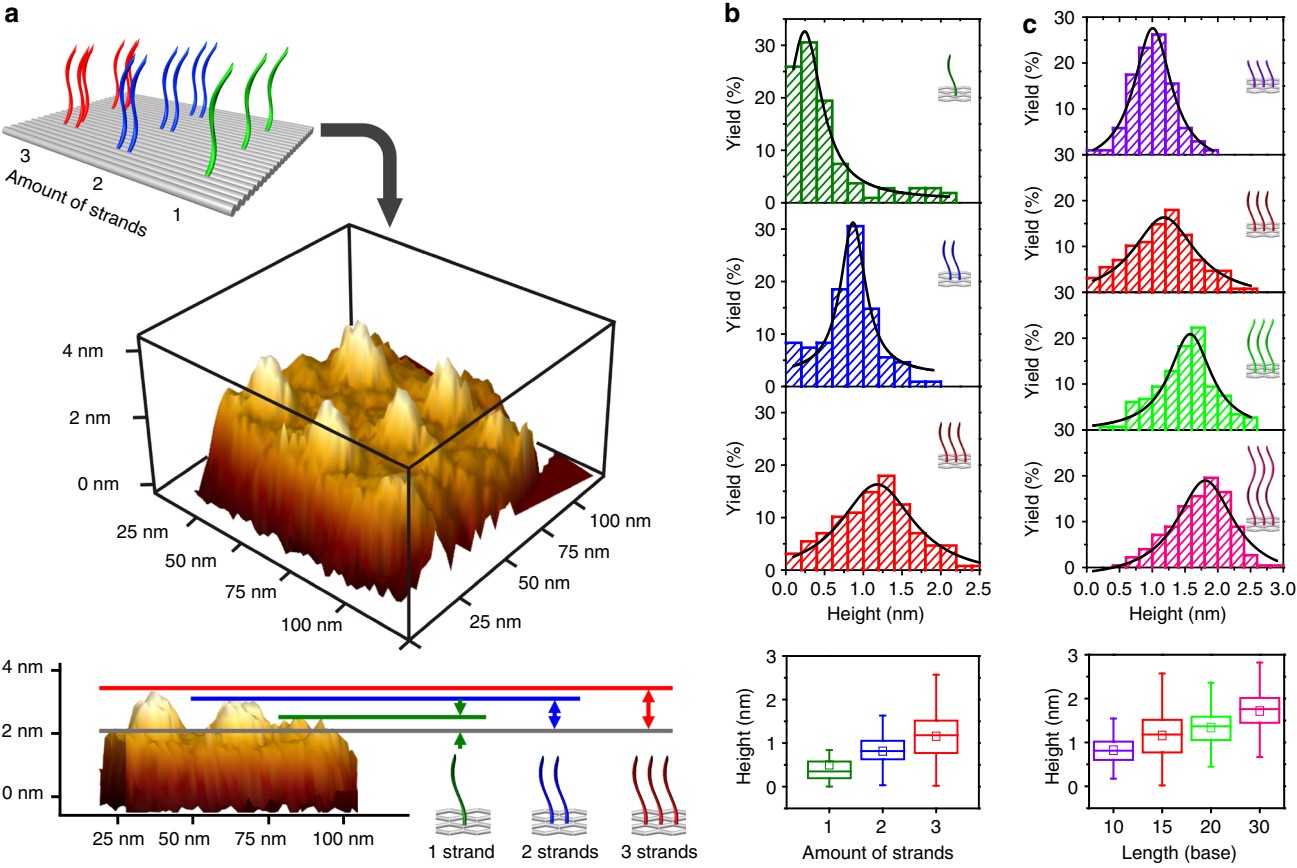

**Fig. 2** Number and length effects of pcDNA on DCIMP. **a** AFM 3D reconstruction (top) and side view (bottom) of a metalized rectangular origami having eight discrete sites with varied numbers of 15-base pcDNA strands. **b** Distribution of increased height values and statistic analysis of height increase on sites with varied number of strands. The error bars are the standard deviation for $N = 108$, 108 and 128 samples with different number of strands. **c** Distribution of increased height values and statistic analysis of height increase on sites with varied length of pcDNA strands. The error bars are the standard deviation for $N = 103$, 128, 147 and 225 samples with different length of pcDNA strands.

metallization. We designed a rectangular origami with eight discrete sites (Supplementary Fig. 23), each carrying pcDNA strands with varied numbers and length (Supplementary Fig. 8a). Using 15-base pcDNA as the model, we found that one-strand sites did not show significant metallization, as revealed by both AFM imaging and statistical analysis of height increase (Fig. 2a, b and Supplementary Fig. 8c). Significantly, two-strand and three-strand sites consistently exhibited height increase of ~1–1.5 nm under AFM, suggesting that the site-specific copper plating only occurs when two or more pcDNA strands are in the proximity. We also note that the length of pcDNA plays an important role in metallization. By anchoring three-strand pcDNA on each site, we found that metallization was complete only when pcDNA is sufficiently long (20–30 base; Fig. 2c). Of note, the inter-strand distances within the pcDNA clusters are of 6 nm, which do not support the interaction and condensation if the length of pcDNA strands is <6 nm.

To measure the line width and density of metallized patterns, we designed a small-sized digit 8 pattern with a size of ~65 × 18 nm$^2$ (Fig. 3a, Supplementary Figs. 9 and 24). After conducting metal plating, we observed a metallized digit 8 occupying approximately half of the origami substrate. AFM measurement revealed that the metallized line width is ~10 nm according, as calculated from the full width at half maximum. The inter-line distance was measured to be 13 nm and the line density was as high as $4.8 \times 10^3$ m cm$^{-2}$ (summarized in Table 1). The successful fabrication of ultrafine lines with 10-nm width suggests that this

on-origami metal painting method is appropriate for constructing metal nanocircuits with ultrahigh density.

Next, we constructed seven-dot and rectangular block patterns to show that this DCIMP strategy works in both positive and negative modes (Supplementary Figs. 10, 25 and 26). In the former approach, we designed a seven-dot pattern, with six dots forming a hexagon and one dot sitting in the center. Each dot contains three strands of pcDNA with 30 bases. Under AFM, we observed clearly visible seven-dot nanopatterns at prescribed positions after metallization, each having a mean height of ~4 nm from the mica, or ~2 nm from the origami. The yield of on-origami Cu metallization reached ~96%, suggesting the high efficiency of this metal-seed-free strategy (Supplementary Fig. 11). Along an alternative approach, we designed a rectangular block pattern for negative-mode painting by removing 20 staple strands in the center of a tall rectangle origami. Single-stranded scaffold segments served as pcDNA strands that were exposed in the hole (22 × 30 nm$^2$). Following the same Cu plating protocol, we painted the hole area with metal without affecting nonvacant part of the origami. Further, we fabricated a three-dot pattern in the hole of a triangle origami that exposed pcDNA at the inner points (Supplementary Fig. 27), with a metallization yield of ~90% (Supplementary Fig. 12).

The nanomechanical properties of the metallized region were quantitatively examined by force-distance (FD) curve-based AFM (FD-based AFM) (Fig. 3b). In FD curves of approach (blue line) and retraction (red line), we observed the coincidence in the

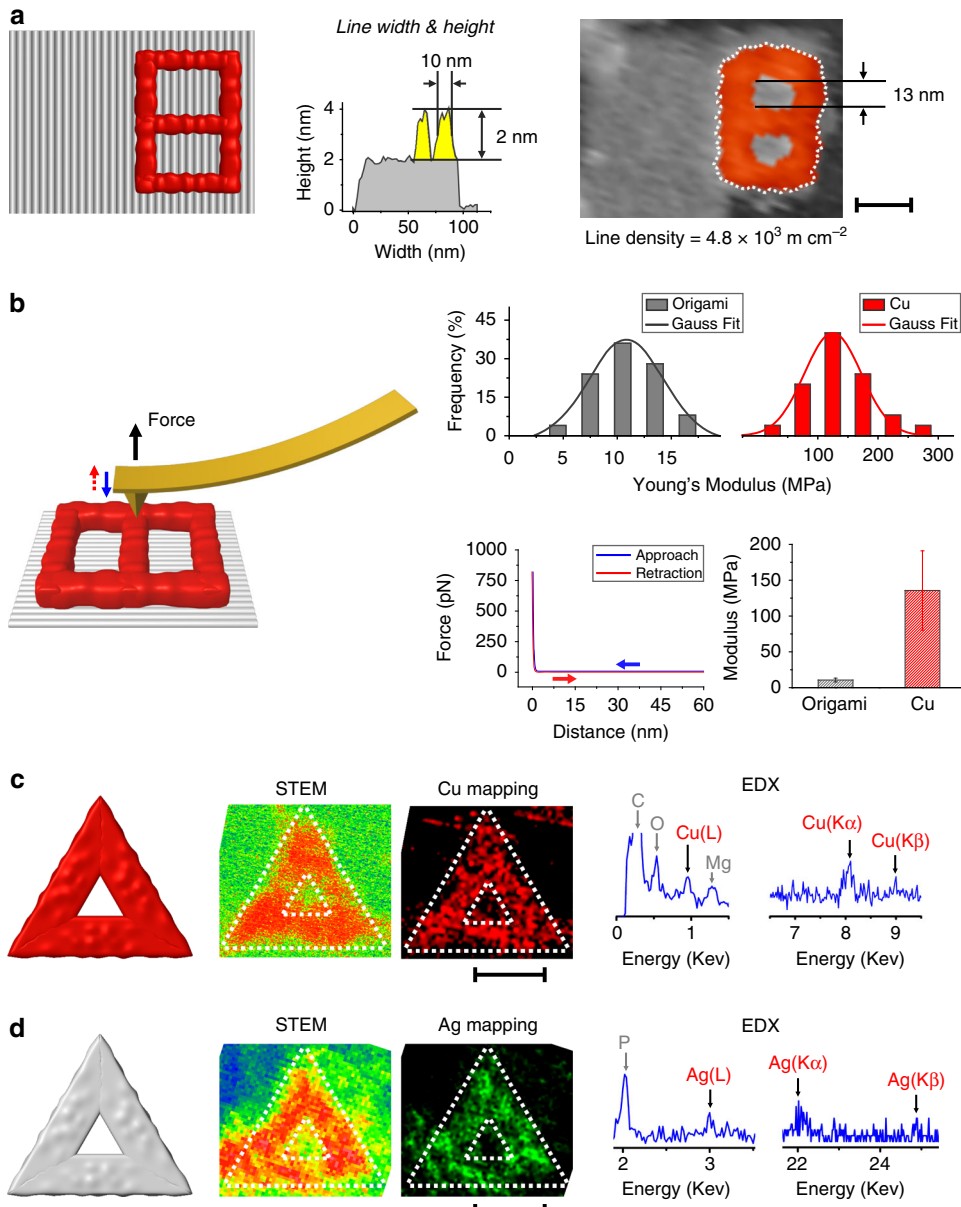

**Fig. 3** Characterization of DCIMP. **a** A small-sized digit 8 pattern enabling the measurement of line width and density. Scale bar: 25 nm. **b** Force-distance (FD) curve-based AFM (FD-based AFM) characterization of nanomechanical properties of metalized patterns. The error bars are the standard deviation for $N = 25$ and 25 samples with the origami and the metalized patterns, respectively. **c** Cu plating on a whole triangular origami with STEM and elemental mapping characterization. Scale bar: 50 nm. **d** Ag plating on a whole triangular origami with STEM and elemental mapping characterization. Scale bar: 50 nm.

**Table 1 Geometric parameters of fabricated nanopatterns.**

| Parameter | Value |
|---|---|
| Line width | 10 nm |
| Height | 2 nm |
| Inter-line distance | 13 nm |
| Line density | $4.8 \times 10^3$ m cm$^{-2}$ |

tested range and a relatively smooth transition around the contact area. The loss of a force curve hysteresis suggested that the metalized patterns were nonviscoelastic. Statistical analysis of extracted Young's modulus revealed that the stiffness increased by more than one order of magnitude after metallization (origami: $10.7 \pm 2.6$ MPa, $N = 25$; metalized patterns: $135.7 \pm 55.4$ MPa, $N = 25$).

To substantiate the Cu nature of metalized patterns, we employed electron microscopy to characterize the metallization process. A fully metalized triangle with pcDNA on all three edges (Supplementary Fig. 28) was characterized with transmission electron microscopy (TEM), scanning transmission electron microscopy (STEM) and energy-dispersive X-ray spectroscopy (EDX) (Fig. 3c and Supplementary Fig. 13). Both EDX point scan and elemental mapping confirmed the Cu nature of metalized patterns. We also imaged a triangular origami with one edge (~20 nm in width) metalized using scanning electron microscopy (SEM). The pcDNA strands in the edge were elongated to create

thick patterns for SEM imaging (Supplementary Figs. 14, 29), whereas no prior staining of the origami was performed. The appearance of bright spots of unstained origami under SEM suggested that the formed metallic layer on origami was conductive.

To further reveal the composition of the metallized product, X-ray photoelectron spectroscopy (XPS) measurements were carried out and the data were analyzed with peak fitting program XPSPEAK (Supplementary Fig. 15a). For Cu metallization, the contents of three species Cu(0), Cu(I) and Cu(II) were 55.2%, 4.0% and 40.9%, respectively, which confirms that the main composition is Cu(0). The presence of Cu(I) and Cu(II) may be caused by incomplete reduction and/or re-oxidation of Cu(0) in air. By comparing the Cu(0) and P contents, the atomic ratio of Cu(0) atoms per pcDNA base was calculated to be 23.4:1.

Having substantiated selective on-origami metallization using bivalent copper ions, we asked whether DCIMP with monovalent silver ions would be possible. We employed $AgNO_3$ and $NH_2OH$ as the Ag source and the reducing agent, respectively. We similarly prescribed a digit 8 pattern on origami with pcDNA strands. After Ag plating, we observed a digit 8 pattern with a clear contour and ~2 nm height increase under AFM (Supplementary Fig. 16). A silver metallized triangular origami was employed for elemental mapping characterization, based on which the composition of Ag element was confirmed (Fig. 3d). The composition determined with XPS revealed that Ag(0) is dominant in the product (86% Ag(0), 14% Ag(1), see Supplementary Fig. 15b). The atomic ratio of Ag(0) atoms per pcDNA base was calculated to be 47.7:1.

**Nano-PCB mimics fabrications.** We next employed DCIMP to construct customized metallic nano-PCB mimics by prescribing a seven-segment digit 8 pattern on origami (Fig. 4a and Supplementary Fig. 17). Each segment was made of a line of pcDNA strands. The combination of various segments supports fabricating 128 nanopatterns including digits and alphabets on the origami canvas. As an example, in order to fabricate an alphabet d pattern, pcDNA was placed on selected segments *B, C, D, E* and *F*. Following the electroless Cu plating reaction, AFM imaging revealed the construction of an expected alphabet d patterned nano-PCB mimic. We further demonstrated the generality of DCIMP by fabricating 15 patterns including digits from 0 to 9 and alphabets of d, n, A and C, u (Fig. 4a, Supplementary Figs. 18, 19).

We further explore the fabrication of double-layer nano-PCB mimics using both Cu and Ag metallization. By exploiting the length-dependent condensation ability, we developed a sequential DCIMP strategy to realize metal ion-specific DNA condensation and metallization (Fig. 4b). To test that, we prepared four units of tall rectangle origami ($100\,nm \times 70\,nm$) with plugs (linker strands), each carrying one alphabet letter C, u, A and g (Stage I) (Supplementary Figs. 30–33). We programmed the pcDNA length by placing 8-base short strands at C and u while 30-base long strands at A and g. Next, we plug these units into a tetramer origami with the size of $100 \times 280\,nm^2$ via hybridization of the linker DNA (stage II). At stage III, we painted Ag on A and g patterns via Ag plating on long pcDNA. Importantly, C and u patterns were unperturbed during Ag plating since short pcDNA does not support metallization. At stage IV, we activated short pcDNA at C and u positions by hybridizing them with 38-base

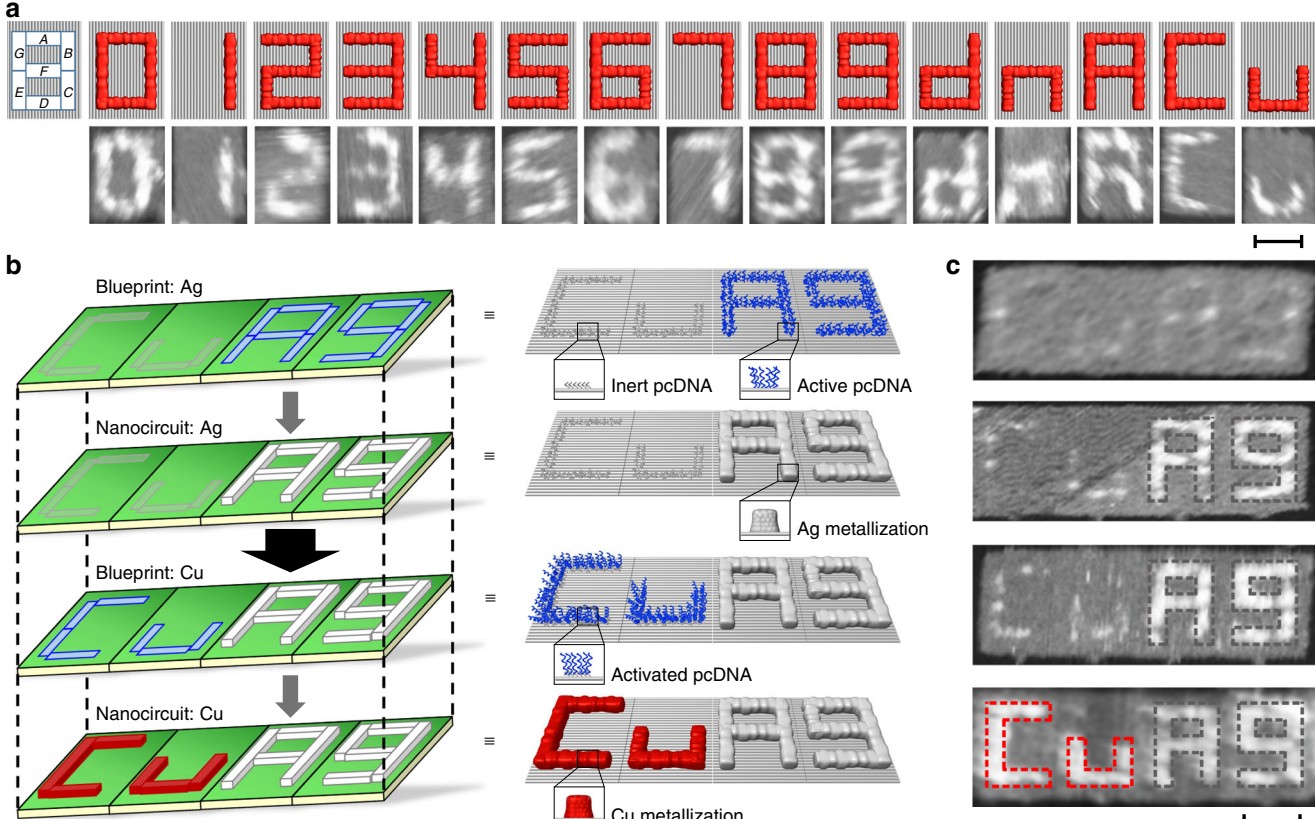

**Fig. 4** Fabricating nano-PCB mimics with DCIMP. **a** A seven-segment digit 8 design is employed to fabricate nano-PCB mimics on a rectangular origami. To show the generality, we fabricated digits from 0 to 9 and several typical alphabets, as shown in corresponding AFM images. **b** Schematic illustration of the fabrication process of Cu-Ag bi-metallic nanocircuits. **c** AFM characterization of the Cu-Ag bi-metallic sample prepared at different stages. The size of the integrated panel is $100 \times 280\,nm^2$. Scale bars: 50 nm.

activation ssDNA handles. The sequence of the 8-base domain was designed to ensure effective hybridization at room temperature (thermodynamic simulation and test are shown in Supplementary Fig. 20). The high density of the handles further promotes the anchoring efficiency. Hybridization of these DNA handles to the C and u positions generated 30-base pcDNA strands that were ready for subsequent Cu plating (Stage V). After processing this sequential DCIMP, we obtained a double-layer nano-PCB mimic that showed bi-metallic patterns of Cu (Cu metal) and Ag (Ag metal) under AFM imaging (Fig. 4c). The EDX measurements confirm the presence of both Cu and Ag elements in the pattern (Supplementary Fig. 21).

## Discussion

In this work, we report the development of DNA origami-enabled selective DNA condensation and metallization reactions. The apparently counterintuitive pcDNA-over-osDNA selectivity arises from their intrinsic structural difference-dependent condensation ability in the presence of low-valence ($z < 3$) metal ions. We find that both the bending stiffness of DNA strands and metal-base coordination play important role in DNA condensation. In the former setting, the bending stiffness of dsDNA excels that of ssDNA by approximately two orders of magnitude[20], and the formation of tightly interconnected origami poses further constraints on osDNA[21]. Thus, given the close relationship between DNA bending stiffness and the condensation efficiency[22], it is reasonable that pcDNA has greater condensation ability than osDNA. In the latter, our theoretical and experimental studies reveal that low-valence metal ions can strongly coordinate with DNA bases that are more exposed in pcDNA than in osDNA which induces attraction and condensation among flexible pcDNA strands.

According to these results, a possible metallization mechanism was proposed. In the original state, pcDNA is placed on prescribed positions to form blueprints for site-specific metallization. The inter-strand distance on each site is fixed to ensure the cation-induced inter-strand condensation. Both the number and the length of pcDNA strands are also found to be critical for high-efficiency metallization. Next, the introduction of metal ions induces the condensation of pcDNA strands on the DNA origami. At this condensed state, metal ions form complexes with DNA bases primarily via N-metal interactions, which increases the local concentration of metal ions. As a result, condensed pcDNA clusters serve as nucleation sites during the metal plating process. The growth of metallized pcDNA clusters and their subsequent connections leads to continuous patterns. Under appropriate metal concentrations, metal plating occurs preferentially on pcDNA over the DNA origami substrate, forming the basis of site-specific metallization. We further note that other mechanisms might also contribute to the observed specificity.

Photolithography plays a key role in modern microelectronic industry. Since the speed and density of integrated circuits are critically dependent on the lithographic resolution, we have seen rapid advances in top-down processing approaches[23]. However, fabricating nanoscale circuits with affordable costs for detection, processing and data storage remains a major challenge[23]. Along a different line, electroless plating-based printed circuit boards (PCBs) represent a cost-effective approach to fabricate industry-scale circuits, which nevertheless has a practical limit of micrometer resolution[24]. The DCIMP strategy for fabricating nano-PCB mimics with high resolution and line density hold great potential for miniaturization of metallic structures toward nanoelectronics and nanophotonics. The all-DNA nature of this DCIMP strategy avoids chemical labeling or metal NP anchoring on DNA. In contrast, previous efforts on on-origami nanolithography generally involve site-specific anchoring of discrete metallic NPs for seeded-growth of metal nano-patterns[18,19,25–28]. In these protocols, the error-prone NP anchoring step, limited hybridization efficiency and surface steric effects often pose limits on geometrical parameters of resultant nanocircuits[1,2,29].

The proof-of-concept demonstration of DCIMP provides a versatile yet precise approach to fabricate ultrafine metal nanostructures with the potential for photonic and electronic applications[15–17,30]. However, the current version of DCIMP requires extensive optimization and generalization for these purposes. First, the development of a generic method for selective metal plating beyond Cu and Ag is indispensable. Second, mass production of DCIMP-based metal nanostructures requires direct metallization in solution, which overcome the constraints of solid substrates. Third, the upgrade from 2D to 3D fabrications is important for exquisite metal fabrication at the nanoscale.

## Methods

**Materials**. All staple strands were purchased from Invitrogen (China) with PAGE purified (sequences listed in the supplementary information). Single-stranded M13mp18 DNA and terminal deoxynucleotidyl transferase (TdT) was purchased from New England Biolabs. Chemicals were purchased from Sinopharm and Sigma-Aldrich.

**Preparation of DNA origami nanostructures**. Single-stranded M13mp18 DNA and staple strands with a molar ratio of 1:10 were mixed in 1× TAE-Mg$^{2+}$ buffer (tris, 40 mM; acetic acid, 20 mM; EDTA, 2 mM; and magnesium acetate, 12.5 mM; pH 8.0). Then they were annealed from 95 to 5 °C for 90 min using a PTC-200 Peltier Thermal Cycler (MJ Research).

**Cu and Ag metallization on DNA origami**. A droplet (6 μL) of DNA origami solution was deposited onto the new cleaved mica for 2 min. Then the surface was rinsed using 1× TA-Mg$^{2+}$ buffer (tris, 40 mM; acetic acid, 20 mM; and magnesium acetate, 12.5 mM; pH 8.0) for six times to get rid of the excess staple strands and EDTA. For the electroless Cu plating, 200 μL of the freshly prepared reducing solution consisting of copper chloride (2–10 mM) and ascorbic acid (20 mM) in 1× TA-Mg$^{2+}$ buffer was added quickly onto the mica surface. The reaction was kept at RT for 10 min and then the Cu plating solution was removed and the mica surface was rinsed with 1× TA-Mg$^{2+}$ buffer.

For Ag metallization, the plating method was similar to the Cu metallization except that AgNO$_3$ and NH$_2$OH were employed as the Ag source and the reducing agent, respectively.

To prepare double-metal DNA origami template, four different letters, C, u, A and g of DNA origami were designed and made separately. Then they were divided into two groups, C and u, A and g, and were mixed equally in molar ratio and were annealed for a second time after purification to remove the excess staple strands and adding new staple strands (L7r2f, L7r6f, L7r10f, L7r14f, L7r18f, L7r22f, L7r26f, L7r30j, L-7r2i, L-7r6e, L-7r10e, L-7r14e, L-7r18e, L-7r22e, L-7r26e, L-7r30e). After that, both were mixed together in equal molar ratio adding another new staple strands (L7r4f, L7r8f, L7r12f, L7r16f, L7r20f, L7r24f, L7r28f, L-7r4e, L-7r8e, L-7r12e, L-7r16e, L-7r20e, L-7r24e, L-7r28e) and were annealed for the third time to form the four-letter pattern, or the double-metal template. The DNA origami was adsorbed onto fresh mica for the plating. The first metal growth on the DNA origami was silver following the same procedure aforementioned. After removing the reaction solution from the surface and washed for three times using 1× TA-Mg$^{2+}$ buffer, the DNA strand (S$_{Cu}$) to hybridize at the "C" and "u" positions on the DNA origami was added onto the surface with a concentration of 1 μM in 1× TA-Mg$^{2+}$ buffer and kept for 1 h at room temperature. Then the solution was removed and the sample was washed for six times using 1× TA-Mg$^{2+}$ buffer. The second plating metal was copper and the same method was used. After reaction the plating solution was removed and the surface was rinsed with 1× TA-Mg$^{2+}$ buffer for three times.

**TdT-catalyzed extension of the staple strands**. All 65 staple strands (1.54 μM for each strand and 100 μM in total, 2.6 μL) for the B side of the triangular origami was mixed in one-pot with dATP (100 mM, 5.2 μL), TdT (20 unit μL$^{-1}$, 32.2 μL), 10× TdT buffer (5 μL) and CoCl$_2$ (2.5 mM, 5 μL) at 37 °C for 1.5 h. The reaction was stopped by heating to 70 °C for 10 min.

**Atomic force microscopy measurements**. All atomic force microscopy (AFM) images were obtained using a Multimode Nanoscope VIII instrument (Bruker) under tapping mode in fluid with SNL-10 tips (Bruker).

The nanomechanical properties were measured with the Multimode Nanoscope VIII instrument (Bruker) in the "PeakForce QNM in Fluid" mode.

"SCANASYST-FLUID+" model cantilever with nominal spring constants of ~0.7 N m$^{-1}$ and tip radius of ~2 nm was chosen (Bruker) for the measurements. In FD-based AFM, the AFM probe was made to approach to and retract from samples to record a series of FD curves. The "NanoScope Analysis" software was used to process the FD curves. The statistical analysis of the Young's modulus distribution was evaluated with about 25 FD curves of origami and Cu metallization pattern, and fitted with a Gaussian model.

**Scanning electron microscopy measurements**. For SEM measurements, plasma-cleaned silicon substrates were used instead of mica. Silicon substrates were cleaned by a Harrick Plasma PDC-32G cleaner for 2 min at high RF level to make the surface hydrophilic. The triangular DNA origami with extended poly-dA overhangs on one side was deposited onto a plasma-cleaned silicon substrate. After Cu plating using aforementioned protocol, the sample was rinsed using water for three times and dried gently under a stream of nitrogen. The sample was scanned by a Hitachi S-4800.

**Transmission electron microscopy measurements**. For TEM measurements, the carbon-coated molybdenum grids (200 mesh, Beijing Zhongjingkeyi Technology Co., Ltd.) were firstly cleaned by a Harrick Plasma PDC-32G cleaner for 30 s at low RF level to make the surface hydrophilic. The purified triangular DNA origami with extended overhangs on three sides was deposited onto a pretreated grid for 2 min. The excess sample was removed away using a piece of filter paper from its edge. After Cu or Ag plating using aforementioned protocol, the plating solution was removed with a filter paper and the grid surface was rinsed with water. The sample was dried at room temperature. The sample was analyzed using an FEI Tecnai G2 F20 S-TWIN operated at 200 kV and equipped with a EDAX Analyser (DPP-II) EDS system and an HAADF (high-angle annular dark field) scanning TEM (STEM) detector.

**X-ray photoelectron spectroscopy measurements**. For XPS measurements, plasma-cleaned silicon substrates were used. Silicon substrates were cleaned by a Harrick Plasma PDC-32G cleaner for 2 min at high RF level to make the surface hydrophilic. The purified triangular DNA origami with extended overhangs on three sides was deposited onto a cleaned silicon substrate for 2 min. The excess sample was removed away using a piece of filter paper from its edge. After Cu or Ag plating using aforementioned protocol, the plating solution was removed with a filter paper and the surface was rinsed with water. The sample was dried at room temperature. The samples were analyzed using a Kratos Axis ULTRADLD system with an Al Kα mono X-ray source of 1486.69 eV at a take-off angle of 54.7°. All the peaks were referenced to the C1s hydrocarbon peak at 285 eV. The data were analyzed using XPSPEAK 4.1 software with Shirley background subtractions and Gaussian-Lorentzian functions for peak fittings.

**Calculation of the theoretical metalized area proportion**. For calculating the theoretical proportion of the metalized digit 8 on the origami substrate, two kinds of methods were used: (1) considering a rectangle origami has 216 staples in total and 75 of them are used for metallization; therefore, the metalized proportion should be 34.72% (Eq. 1); (2) considering the size of a rectangle origami is 100 nm × 70 nm and the metalized area has a size of 2813 nm$^2$, therefore, the metalized proportion should be 40.19% (Eq. 2). Furthermore, assuming that the errors in AFM measurement are ~10%, the theoretical proportion of the metalized area should be in the range of 31.25–44.21%.

$$\text{metallized proportion} = \frac{\text{number of pcDNA}}{\text{number of total staples}} = \frac{75}{216} = 34.72\%, \quad (1)$$

$$\text{metallized proportion} = \frac{\text{size of metalized area}}{\text{size of origami substrate}} = \frac{2813\,\text{nm}^2}{7000\,\text{nm}^2} = 40.19\%. \quad (2)$$

**Theoretical simulation and calculations**. Molecular dynamic (MD) simulation was performed by Gromacs[31] to observe the interactions between ions and DNA origami in the explicit solvent. The seven-dot pattern DNA origami was prepared with the parmbsc0 refined amber FF[32] and solvated in a cubic box of TIP3P water molecules[33]. The size of the solvent box was 12 × 14 × 14 nm$^3$. 180 Mg$^{2+}$ and 11 Cl$^-$ ions were added to neutralize the system. We employed a new 12–6–4 LJ-type model taking into account the ion-induced dipole interaction to calculate the Van der Waals contributions between cations and nucleotides[34]. The particle mesh Ewald (PME) method[35] was used to calculate the Coulombic interactions. The cutoff value of nonbonded interactions was set to 1.5 nm. The SHAKE algorithm[36] was used to restrain all of the bond lengths that involved hydrogen atoms with a 2-fs time step. To observe the movement of cations and pcDNA well, a week force constant of 1000 kJ mol$^{-1}$ nm$^{-2}$ as a harmonic constraint was applied on osDNA during the MD simulation. After a 4-ns MD simulation, 30 Cu$^{2+}$ and 60 Cl$^-$ ions were randomly added into the system. Limited by the spatial scale, the concentration of the cations in the MD simulation could not be corresponded with the experiment. As the optimal Cu$^{2+}$ concentration was 2 mM in the experiment, which was one sixth of the Mg$^{2+}$ concentration, we added 30 Cu$^{2+}$ ions into the solvent by making reference to the percentage. An extra 10-ns MD simulation was

performed under the same conditions. The numbers of Mg$^{2+}$ and Cu$^{2+}$ cations near 15 Å (equal to the cutoff value of nonbonded interactions) of the pcDNA and osDNA were counted during the MD simulations, respectively. The radial distribution function (RDF) of the cations around the phosphate oxygen atoms was calculated by the g_rdf command.

For quantum mechanical calculations, the initial complexes were created by placing the hydrated metal cations near the N7 position of guanine or N3 position of cytosine or near the phosphate oxygens. Geometry optimizations were carried out with B3LYP method[37] and the 6–311++G** basis set and the solvent effects were modeled by the CPCM (COSMO) Model[38]. The LANL2DZ relativistic pseudopotential and basis[39] was used for Cu$^{2+}$ ion. Frequency calculations were performed to verify the energy minima. Interaction energies of the hydrated metal ions and the base were estimated as follows:[40]

$$\Delta E = (E_{\text{X-M-5W}} + E_{\text{W}}) - (E_{\text{X}} + E_{\text{M-6W}}), \quad (3)$$

where $E_{\text{X-M-5W}}$, $E_{\text{W}}$, $E_{\text{X}}$ and $E_{\text{M-6W}}$ are the respective energies of the base-pentahydrated metal ion complex (X = G, C, A and M = Mg$^{2+}$, Cu$^{2+}$), single water molecule, base, and the hexahydrated metal ion. Similarly, interaction energies of the hydrated metal ions and the phosphate group were determined as follows:

$$\Delta E = E_{\text{P-M-6W}} - (E_{\text{P}} + E_{\text{M-6W}}), \quad (4)$$

where $E_{\text{P-M-6W}}$ and $E_{\text{P}}$ are the respective energies of the phosphate group-hexahydrated metal ion complex and phosphate group. All calculations were carried out using the Gaussian 09 program (Gaussian, Inc., Wallingford CT, 2009).

## Data availability
The data presented in this paper are available from the corresponding author upon reasonable request. The source data underlying Figs. 1d, 2b, c, 3b and Supplementary Fig. 20b are provided as a Source Data file.

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

## Acknowledgements

We greatly appreciate the financial support from the Ministry of Science and Technology of China (2016YFA0400900), National Science Foundation of China (21834007, 21873071, 9427304, 21473236, 21722310, 11374221, 11574224, 61378062), the Innovative research team of high-level local universities in Shanghai, the Key Research Program of Frontier Sciences (QYZDJ-SSW-SLH031), the Open Large Infrastructure Research of CAS, the LU Jiaxi International Team of the Chinese Academy of Sciences, and the K.C. Wong Foundation at Shanghai Jiao Tong University.

## Author contributions

C.F. and H.L. supervised the research. C.F, H.L., S.J. and J.W. designed the experiments. S.J., J.W. and M.X. performed the experiments and analyzed the data. J.S. and J.Lin did the quantum mechanical calculation. Y.Z., J.C., J.Li., L.W. and K.V.G. discussed the results. S.J., J.W., H.L. and C.F. interpreted the data and wrote the manuscript, and all authors read and commented on the manuscript.

## Competing interests

The authors declare no competing interests.
