## [Peer Review File · Nature Communications]

Reviewers' comments:

Reviewer #1 (Remarks to the Author):

The paper of Fan et al. describes DNA based metallization process for preparation of metal nanostructures with a nanometer precision. The method is interesting and robust and technical details are well described. However, I find insufficient the explanation on the mechanism and products of the metallization and I am concerned about very imprecise discussion related to DNA condensation that should be revised. The major points of my concern are summarized below.

Major points

1. Authors do not explain how the metallization of the template happens and what is the composition and the structure of the product. Does Cu^{2+} (Ag^{+}) ion forms a complex with DNA and induce its condensation? After reduction of Cu^{2+} , in which form is the metal phase generated, metal clusters, nanoparticles or none of these? What is the composition of the metallization product in (Cu atoms per DNA base pair)? Metallization scenario must be explained and it would be nice to have schematics.
2. The lack of knowledge about "DNA condensation" field of the authors may cause some misunderstand of a part of the paper and will be probably confusing for polymer physics scientists. The places below contain questionable statements related to DNA condensation.
(lines 62-67) This part in introduction on DNA condensation is somewhat irrelevant to the preformed study as natural DNA condensation as well as multivalent condensation theory are about double-stranded single DNA macromolecules, whereas authors describe experiments with single stranded DNA or origami. The references 1 and 2 are not general enough to represent DNA condensation studies in vivo.
(lines 97-100) An explanation included like-charge attraction is rather confusing. Thinking only about like-charge interactions, interaction of Cu^{2+} with osDNA and pcDNA should also promote pcDNA adsorption to osDNA, but in this case no metallization selectivity can be expected. To reviewer, the reason of selectively is the difference between ss and ds-DNA interaction with Cu^{2+} . Cu^{2+} presumably interacts with DNA bases and thus Cu^{2+} becomes "hidden" inside dsDNA helix while outer phosphates would still be anionic and, unless Cu^{2+} is added in a large excess, there would be no cation-induced condensation. On the other hand, bases of ssDNA are exposed to solution and Cu^{2+} can efficiently form bridges between two neighboring pcDNA strands promoting their condensation.
(lines 293-294) The statement about selective metallization due to "intrinsic condensation ability" in the presence of low-valence metal ions is difficult to understand. Generally speaking, it is not correct to compare directly origami and single chain of ssDNA – they are two structurally different DNA existences. Second, even there might be some relation between Cu^{2+} induced pcDNA condensation and metallization, how can osDNA condensation promote metallization?
(lines 302-304) It is a misconception about $z < 3$ ions that cannot condense DNA and reference used in this contents. The theory of counterion condensation, demonstrated that $z < 3$ ions that cannot condense DNA, was developed for double stranded DNA and for ions that interact purely electrostatically with DNA backbone. To the best of my knowledge, theories for condensation of ssDNA or origami were not elaborated, and, in any case, they should be quite different from classic dsDNA condensation model. To say more, the studied ions can induce DNA condensation via other ways rather than electrostatic interaction, by, for example, strongly decreasing DNA persistence length.
(304-307) This sentence is unclear. What kind of physiologically relevant process do they mean? If it is DNA condensation, then the system they studied has little to do with physiological process because no heavy metals in cells can be found at such amounts to induce DNA condensation.
3. (line 170-171) The statement about "catalytically induced Cu metallization and growth" is not correct. By definition, catalyst is not consumed and can be used repeatedly. In the present study, however, pcDNA cannot be regenerated, thus the role of pcDNA should be acknowledged as template (nucleation center) but not as catalyst.
4. (lines 183-191) More explanation on the role of pcDNA length in nanostructures formation must be provided. Why is the length of pcDNA important? The authors should make unambiguously clear the conditions for effective metallization. Does the length of pcDNA longer than the distance between

adjacent pcDNA is a key for successful interchain interaction to form condensed metallized nanostructure? Or condensation of DNA occurs at the level of single pcDNA chains and thus multiple intrachain interactions become important?

5. (323-326) This sentence is too pretentious. In the paper, no intrinsic DNA condensation properties were exploited. In the same sentence, the phrase "site-specific metallization" can be incorrectly interpreted as it can be read as DNA sequence specific metallization, which is not what they mean. The phrase "precise approach with desirable optical and electronic properties" claims the achievements that were not demonstrate in paper, especially, about optical properties.

Minor

I found it difficult to read the paper as I always had to refer to supporting information, which contains quite a number of important data.

Reviewer #2 (Remarks to the Author):

The manuscript by Jia et al reported a surprisingly efficient label-free nanofabrication method that metalizes single-stranded DNA clusters (pcDNA) displayed on DNA-origami sheets with high specificity in a one-step reaction. Low-valence cations, including Cu(II) and Ag(I), has been reduced in a site-specific way on pcDNA, generating programmable metalized patterns with fine spatial control and considerable design flexibility. While most characterization were done through AFM, electron microscopy and EDX techniques were used to confirm the quality of the "nano-PCB" products. The problem is important, the method is elegant, and the data quality is high. I support publication after addressing my points as follows.

My major concerns are with the proposed mechanism of metalization and potential limitations of the technique.

1. About the metalization mechanism, it is not clear to me at which stage the "nucleation" happens. I.e., do the unreduced cations condense the pcDNA clusters (as proposed by the authors) or do the pcDNA clusters bind to the metal colloids at early stage of the reduction? If I read correctly, DNA condensation was never directly observed in this study. Additionally, while it is helpful to know the delta-G's for metal-base binding, one should also keep in mind the electrostatic interactions between cations and the DNA phosphate backbones. Given that the method has been empirically tuned to work beautifully for the purpose outlined here (metallization patterning with 10-nm resolution), I do not expect a full mechanistic investigation. But a fair and square discussion is in order.
2. The authors may also want to discuss the limitations of the technique. For example, in this study the metalization is done on surface, allowing convenient washing-away of reactants and non-specific products. Is this technique applicable in solution, potentially for high-volume, mass production? Besides Ag and Cu, what other types of low-valence cations are amenable to this fabrication method? Is the nano-patterning limited to 2D (DNA-origami structures are certainly not)?

Minor points:

1. Authors described the rectangle DNA origami as "stiff" and "perfect 2D crystal". While I realize that it's all relative, generally one would not use these descriptors on the single-layer DNA-origami sheets, especially considering the experimental evidence of self-curling and surface defects.
2. It helps to point out the inter-strand distance within the pcDNA site and the inter-cluster distance between the pcDNA sites.
3. Authors wrote "the stiffness of a given object is proportional to the forth order of its effective radius". I know this is true for bending stiffness (and perhaps that what the authors meant here), but I am not sure if this applies to other mechanical stiffness (e.g. shearing stiffness).
4. In the sentence "we reason that metal ions-induced compensation of the negative ion atmosphere of DNA". Do you mean "compensation" or "condensation"?
5. For clarity, it will be helpful to define "occupancy" in Fig.1 in addition to in the main text, perhaps

by moving some of the tomography images in Fig. S6 to Fig. 1.

6. The statement "Mg²⁺ ions are indispensable for the origami assembly" is not true. Others have demonstrated that other ions, both single valence and multi valence, can facilitate DNA-origami folding in place of Mg.

7. In my opinion, Fig 4a and 4b have low information density: they illustrate quite obvious concepts in unnecessarily complex ways. I recommend replacing them with the multi-step fabrication scheme in Fig. S18, which, to me, is far more interesting and informative.

8. The pseudo-color in Fig 4d needs to be explained.

9. For sequential metalization, 8-nt pcDNAs were hybridized to 38-nt ssDNA handles. 8-bp dsDNA is thermodynamically unstable at room temperature. It worked here possibly due to the high density of the probes. It may be worth pointing out.

10. It would be nice to have EDX data on the Cu-Ag hybrid pattern. What is the fabrication yield of this pattern (multi-step reactions are usually associated with lower yield)?

Responses to the reviewers:

Reviewer #1

The paper of Fan et al. describes DNA based metallization process for preparation of metal nanostructures with a nanometer precision. The method is interesting and robust and technical details are well described. However, I find insufficient the explanation on the mechanism and products of the metallization and I am concerned about very imprecise discussion related to DNA condensation that should be revised. The major points of my concern are summarized below.

Reply: We are grateful for the reviewer's very positive comments.

Major points

1. Authors do not explain how the metallization of the template happens and what is the composition and the structure of the product. Does Cu^{2+} (Ag^+) ion forms a complex with DNA and induce its condensation? After reduction of Cu^{2+} , in which form is the metal phase generated, metal clusters, nanoparticles or none of these? What is the composition of the metallization product in (Cu atoms per DNA base pair)? Metallization scenario must be explained and it would be nice to have schematics.

Reply: We highly appreciate the constructive suggestion from this reviewer.

Based on our quantum mechanical calculations, Cu^{2+} ions strongly coordinated with DNA bases via the N-Cu interactions, indicating the formation of Cu-DNA complex (Fig. 1b). When there are sufficient numbers of pcDNA strands on a single site, interactions between neighboring Cu-DNA complexes can induce condensation for subsequent metal plating, resulting in the formation of Cu nanoparticles. We further studied the effects of pcDNA number and length on this process (Fig. 2).

In order to reveal the composition of the product, we carried out new XPS measurements (Fig. S15). For Cu metallization, the contents of three species Cu(0), Cu(I) and Cu(II) were 55.2%, 4.0% and 40.9%, respectively, which confirms that the main composition is Cu(0). The presence of Cu(I) and Cu(II) may be caused by incomplete reduction and/or re-oxidation of Cu(0) in air. For Ag metallization, XPS revealed that only 14% of Ag(I) was found, whereas 86% of the component was Ag(0). Based on XPS quantitative elemental analysis, the calculated metal atoms per DNA base (pcDNA) in the metallization product were 23.4:1 (Cu) and 47.7:1 (Ag).

We have provided the new data in the revised version. Also as suggested, a new scheme depicting the metallization process was added in the Supplementary Information (Fig. S1).

2. The lack of knowledge about "DNA condensation" field of the authors may cause some misunderstand of a part of the paper and will be probably confusing for polymer physics scientists. The places below contain questionable statements related to DNA condensation.

(lines 62-67) This part in introduction on DNA condensation is somewhat irrelevant to

the preformed study as natural DNA condensation as well as multivalent condensation theory are about double-stranded single DNA macromolecules, whereas authors describe experiments with single stranded DNA or origami. The references 1 and 2 are not general enough to represent DNA condensation studies in vivo.

Reply: We agree with the reviewer that the condensation in our study is different from that happens in vivo. We were in fact inspired by in-vivo condensation that cations induce negative DNA chains to form compact structure, which in turn cooperatively increases the local concentration of cations. We thus modified the description in the text (Lines 66-69, Page 3). We also replaced the references 1 and 2 with new ones that review the general process of DNA condensation in vivo.

(lines 97-100) An explanation included like-charge attraction is rather confusing. Thinking only about like-charge interactions, interaction of Cu^{2+} with osDNA and pcDNA should also promote pcDNA adsorption to osDNA, but in this case no metallization selectivity can be expected. To reviewer, the reason of selectively is the difference between ss and ds-DNA interaction with Cu^{2+} . Cu^{2+} presumably interacts with DNA bases and thus Cu^{2+} becomes “hidden” inside dsDNA helix while outer phosphates would still be anionic and, unless Cu^{2+} is added in a large excess, there would be no cation-induced condensation. On the other hand, bases of ssDNA are exposed to solution and Cu^{2+} can efficiently form bridges between two neighboring pcDNA strands promoting their condensation.

Reply: We agree with the reviewer that the term “like-charge interaction” may cause confusion. We also agree with the reviewer that the metallization selectivity is probably due to the difference in affinities to metal ions of ss- and ds-DNA. From our quantum mechanical calculations, Cu^{2+} ions tended to coordinate with DNA bases while phosphates were neutralized with Mg^{2+} ions. Therefore cation-induced condensation occurred on ssDNA (pcDNA) sites. The term “like-charge interaction” has been removed and the text has been modified.

(lines 293-294) The statement about selective metallization due to “intrinsic condensation ability” in the presence of low-valence metal ions is difficult to understand. Generally speaking, it is not correct to compare directly origami and single chain of ssDNA – they are two structurally different DNA existences. Second, even there might be some relation between Cu^{2+} induced pcDNA condensation and metallization, how can osDNA condensation promote metallization?

Reply: We accept this comment with appreciation. Indeed, we consider that the selectivity of the metallization is from the different states of the pcDNA and the osDNA molecules. The pcDNA molecules protrude into the solution and the bases bind to the cation ions, which induce the condensation of the pcDNA and promote the metallization. In contrast, the high stiffness and large persistence length of osDNA prevent their compaction even in the presence of cations. We provided new explanations in the

revised manuscript (Lines 328-342, Pages 17-18).

(lines 302-304) It is a misconception about $z < 3$ ions that cannot condense DNA and reference used in this contents. The theory of counterion condensation, demonstrated that $z < 3$ ions that cannot condense DNA, was developed for double stranded DNA and for ions that interact purely electrostatically with DNA backbone. To the best of my knowledge, theories for condensation of ssDNA or origami were not elaborated, and, in any case, they should be quite different from classic dsDNA condensation model. To say more, the studied ions can induce DNA condensation via other ways rather than electrostatic interaction, by, for example, strongly decreasing DNA persistence length.

Reply: We are grateful for this professional suggestion. The text about $z < 3$ ions that cannot condense DNA was removed.

(304-307) This sentence is unclear. What kind of physiologically relevant process do they mean? If it is DNA condensation, then the system they studied has little to do with physiological process because no heavy metals in cells can be found at such amounts to induce DNA condensation.

Reply: This comment has been accepted. We agree with the reviewer that physiological process is not strongly relevant to our work. Correction has been made in the revised manuscript.

3. (line 170-171) The statement about “catalytically induced Cu metallization and growth” is not correct. By definition, catalyst is not consumed and can be used repeatedly. In the present study, however, pcDNA cannot be regenerated, thus the role of pcDNA should be acknowledged as template (nucleation center) but not as catalyst.

Reply: We agree that the term “catalyst” was used incorrectly. The text has been corrected in the revised manuscript.

4. (lines 183-191) More explanation on the role of pcDNA length in nanostructures formation must be provided. Why is the length of pcDNA important? The authors should make unambiguously clear the conditions for effective metallization. Does the length of pcDNA longer than the distance between adjacent pcDNA is a key for successful interchain interaction to form condensed metallized nanostructure? Or condensation of DNA occurs at the level of single pcDNA chains and thus multiple intrachain interactions become important?

Reply: We appreciate that the reviewer raised this important issue. The length of the pcDNA is important since it has to be longer than the inter-strand distances to enable condensation between neighboring strands, as the reviewer pointed out. Also, according to our experimental results (Fig. 2), longer pcDNA facilitates the condensation since it promotes the multiple interaction and coordinate condensation in a single pcDNA chain

as well as among adjacent chains. New discussions about the role of the length of the pcDNA has been added in the revised manuscript (Lines 196-198, Page 10).

5. (323-326) This sentence is too pretentious. In the paper, no intrinsic DNA condensation properties were exploited. In the same sentence, the phrase “site-specific metallization” can be incorrectly interpreted as it can be read as DNA sequence specific metallization, which is not what they mean. The phrase “precise approach with desirable optical and electronic properties” claims the achievements that were not demonstrate in paper, especially, about optical properties.

Reply: This constructive suggestion has been followed. These claims have been removed in the revised manuscript.

Minor

I found it difficult to read the paper as I always had to refer to supporting information, which contains quite a number of important data.

Reply: This comment has been accepted. We have re-arranged several figures and modified some text in the revised version.

Reviewer #2

The manuscript by Jia et al reported a surprisingly efficient label-free nanofabrication method that metalizes single-stranded DNA clusters (pcDNA) displayed on DNA-origami sheets with high specificity in a one-step reaction. Low-valence cations, including Cu(II) and Ag(I), has been reduced in a site-specific way on pcDNA, generating programmable metalized patterns with fine spatial control and considerable design flexibility. While most characterization were done through AFM, electron microscopy and EDX techniques were used to confirm the quality of the "nano-PCB" products. The problem is important, the method is elegant, and the data quality is high. I support publication after addressing my points as follows.

Reply: We are thankful that the reviewer appreciates the importance of our work and provides very positive comments.

My major concerns are with the proposed mechanism of metalization and potential limitations of the technique.

1. About the metalization mechanism, it is not clear to me at which stage the "nucleation" happens. I.e., do the unreduced cations condense the pcDNA clusters (as proposed by the authors) or do the pcDNA clusters bind to the metal colloids at early stage of the reduction? If I read correctly, DNA condensation was never directly observed in this study. Additionally, while it is helpful to know the delta-G's for metal-base binding, one should also keep in mind the electrostatic interactions between cations and the DNA phosphate backbones. Given that the method has been empirically tuned to work beautifully for the purpose outlined here (metallization patterning with 10-nm resolution), I do not expect a full mechanistic investigation. But a fair and square discussion is in order.

Reply: We are thankful that the reviewer pointed out this important issue. In the revised manuscript we added a paragraph discussing about it (Lines 328-342, Pages 17-18).

“According to these results, a possible metallization mechanism was proposed. In the original state, pcDNA is placed on prescribed positions to form blueprints for site-specific metallization. The inter-strand distance on each site is fixed to ensure the cation-induced inter-strand condensation. Both the number and the length of pcDNA strands are also found to be critical for high-efficiency metallization. Next, the introduction of metal ions induces the condensation of pcDNA strands on the DNA origami. At this condensed state, metal ions form complexes with DNA bases primarily via N-metal interactions, which increases the local concentration of metal ions. As a result, condensed pcDNA clusters serve as nucleation sites during the metal plating process. The growth of metallized pcDNA clusters and their subsequent connections leads to continuous patterns. Under appropriate metal concentrations, metal plating occurs preferentially on pcDNA over the DNA origami substrate, forming the basis of site-specific metallization. We further note that other mechanisms might also contribute

to the observed specificity. For example, metal nuclei preformed in solution might preferably bind to pcDNA to facilitate site-specific metallization.”

2. The authors may also want to discuss the limitations of the technique. For example, in this study the metalization is done on surface, allowing convenient washing-away of reactants and non-specific products. Is this technique applicable in solution, potentially for high-volume, mass production? Besides Ag and Cu, what other types of low-valence cations are amenable to this fabrication method? Is the nano-patterning limited to 2D (DNA-origami structures are certainly not)?

Reply: We are grateful for the expertise of the reviewer. In the revised manuscript we added a paragraph discussing about it (Lines 359-365, Page 19).

“The proof-of-concept demonstration of DCIMP provides a versatile yet precise approach to fabricate ultrafine metal nanostructures with the potential for photonic and electronic applications. However, the current version of DCIMP requires extensive optimization and generalization for these purposes. First, the development of a generic method for selective metal plating beyond Cu and Ag is indispensable. Second, mass production of DCIMP-based metal nanostructures requires direct metallization in solution, which overcome the constraints of solid substrates. Third, the upgrade from 2D to 3D fabrications is important for exquisite metal fabrication at the nanoscale.”

Minor points

1. Authors described the rectangle DNA origami as "stiff" and "perfect 2D crystal". While I realize that it's all relative, generally one would not use these descriptors on the single-layer DNA-origami sheets, especially considering the experimental evidence of self-curling and surface defects.

Reply: We appreciate this professional comment. We have removed these descriptions in the revised manuscript.

2. It helps to point out the inter-strand distance within the pcDNA site and the inter-cluster distance between the pcDNA sites.

Reply: Thanks for this excellent suggestion. The distances of the inter-strand within the pcDNA site are fixed at 6 nm. The inter-cluster distances between the pcDNA sites are also 6 nm for the patterns containing continuous lines, e.g., the “digital 8”. This information has been added in the revised manuscript (Lines 196-198, Page 10).

3. Authors wrote "the stiffness of a given object is proportional to the forth order of its effective radius". I know this is true for bending stiffness (and perhaps that what the authors meant here), but I am not sure if this applies to other mechanical stiffness (e.g. shearing stiffness).

Reply: We appreciate the expertise of the reviewer. For this statement in the text, indeed we were referring to the bending stiffness. In the revised manuscript, we use the term “bending stiffness”.

4. In the sentence "we reason that metal ions-induced compensation of the negative ion atmosphere of DNA". Do you mean "compensation" or "condensation"?

Reply: We thank the reviewer for pointing out this error, which has been corrected in the revised manuscript.

5. For clarity, it will be helpful to define "occupancy" in Fig.1 in addition to in the main text, perhaps by moving some of the tomography images in Fig. S6 to Fig. 1.

Reply: We appreciate this constructive suggestion. Accordingly, we have defined the “occupancy” in the caption of Fig. 1. The old Fig. S6 was merged into Fig. 1 to illustrate this definition in the revised manuscript.

6. The statement "Mg²⁺ ions are indispensable for the origami assembly" is not true. Others have demonstrated that other ions, both single valence and multi valence, can facilitate DNA-origami folding in place of Mg.

Reply: We thank the reviewer for pointing out this problem. In the revised manuscript, we have corrected this statement.

7. In my opinion, Fig 4a and 4b have low information density: they illustrate quite obvious concepts in unnecessarily complex ways. I recommend replacing them with the multi-step fabrication scheme in Fig. S18, which, to me, is far more interesting and informative.

Reply: We highly appreciate this constructive suggestion. We have replaced Fig. 4a, 4b and 4d with Fig. S18 as the reviewer pointed out. The old Fig. 4a and 4b have been moved to the Supplementary Information (Fig. S17).

8. The pseudo-color in Fig 4d needs to be explained.

Reply: We thank the reviewer for pointing out this problem. This pseudo-color of the Cu-Ag hybrid pattern was set to distinguish them from each other. In the revised manuscript, this figure has been removed and the old Fig. S18 was used here.

9. For sequential metalization, 8-nt pcDNAs were hybridized to 38-nt ssDNA handles. 8-bp dsDNA is thermodynamically unstable at room temperature. It worked here possibly due to the high density of the probes. It may be worth pointing out.

Reply: This suggestion has been followed with thanks. We agree with the reviewer that

the high density of the probes is important for the successful metallization. It is also noted that the sequence of the 8-nt pcDNAs was carefully designed to ensure stable hybridization at room temperature. The simulated and experimental results were provided in the revised manuscript (Fig. S20).

10. It would be nice to have EDX data on the Cu-Ag hybrid pattern. What is the fabrication yield of this pattern (multi-step reactions are usually associated with lower yield)?

Reply: We appreciate the constructive suggestion from this reviewer. As provided in the Supplementary Information (Fig. S21), the EDX measurements confirms the presence of both Cu and Ag elements in the pattern. The fabrication yield is indeed low (~10%) since this pattern was assembled from four tall rectangle units and operated in ten steps.

REVIEWERS' COMMENTS:

Reviewer #1 (Remarks to the Author):

Generally, I am satisfied with the revision performed. I would like to suggest a couple of further (not critical) changes to be considered by authors.

1. While the new reference [1] is appropriate, the reference [2] is not. I suggest you to remove the reference [2] and cite classic work of Bloomfield (Bloomfield, V. A., DNA condensation. *Curr. Opin. Struc. Biol.* 1996, 6 (3), 334-341.)

2. In the revised text, authors have added "For example, metal nuclei preformed in solution might preferably bind to pcDNA to facilitate site-specific metallization". I recommend not to add this sentence because nucleation on a template such as DNA is much more energetically favorable, therefore, of nucleation in solutions would hardly contribute much to the metallization process.

Reviewer #2 (Remarks to the Author):

The authors have adequately addressed my points. The revised manuscript clearly defines the limitations of the method and gives a plausible explanation of the metallization mechanism. The revised figures are more informative and easier to follow. I recommend publication.

Responses to the reviewers:

Reviewer #1

Generally, I am satisfied with the revision performed. I would like to suggest a couple of further (not critical) changes to be considered by authors.

Reply: We are grateful for the reviewer's very positive comments.

1. While the new reference [1] is appropriate, the reference [2] is not. I suggest you to remove the reference [2] and cite classic work of Bloomfield (Bloomfield, V. A., DNA condensation. *Curr. Opin. Struc. Biol.* 1996, 6 (3), 334-341.)

Reply: We would like to thank the reviewer for the suggestion. The reference [2] has been replaced.

2. In the revised text, authors have added "For example, metal nuclei performed in solution might preferably bind to pcDNA to facilitate site-specific metallization". I recommend not to add this sentence because nucleation on a template such as DNA is much more energetically favorable, therefore, of nucleation in solutions would hardly contribute much to the metallization process.

Reply: This constructive suggestion has been followed. That sentence has been removed.

Reviewer #2

The authors have adequately addressed my points. The revised manuscript clearly defines the limitations of the method and gives a plausible explanation of the metallization mechanism. The revised figures are more informative and easier to follow. I recommend publication.

Reply: We are thankful that the reviewer appreciates our work.